# Self-Rated Health and Mortality: Moderation by Purpose in Life

**DOI:** 10.3390/ijerph20126171

**Published:** 2023-06-19

**Authors:** Elliot M. Friedman, Elizabeth Teas

**Affiliations:** Department of Human Development and Family Science, Purdue University, West Lafayette, IN 47907, USA

**Keywords:** self-rated health, purpose in life, mortality, racial differences

## Abstract

Poor self-rated health consistently predicts reduced longevity, even when objective disease conditions and risk factors are considered. Purpose in life is also a reliable predictor of diverse health outcomes, including greater longevity. Given prior work in which we showed that purpose in life moderated the association between chronic conditions and health-related biological factors, the aim of the current study was to examine the role of purpose in life in moderating the relationship between subjective health and mortality. We also examined potential differences in these associations by race/ethnicity. Data were from two large national longitudinal studies—the Health and Retirement Study (HRS) and the Midlife in the United States (MIDUS) study—with a 12- to 14-year follow-up period for mortality estimates. Results of logistic regression analyses showed that purpose in life and self-rated health were both significantly positively associated with longevity, and that purpose in life significantly moderated the relationship between self-rated health and mortality. Stratified analyses showed similar results across all racial/ethnic groups, with the exception of Black MIDUS participants. These results suggest that greater purpose in life may provide a buffer against the greater probability of mortality associated with poor subjective health.

## 1. Introduction

For more than 60 years, studies in multiple countries spanning young, middle, and older adulthood have shown that better subjective health ratings predict a lower risk of mortality [1,2,3]. Self-rated health is largely influenced by objective health status, such as the presence of disease conditions and/or reduced functional capacity, particularly among older adults [3] and those who rate their health worse [4]. Moreover, the power of self-rated health to predict mortality typically improves as people have more accurate information about their health [3]. Nevertheless, the relationship between self-rated health and mortality persists in analyses that account for myriad chronic diseases and risk factors [1,2,3], especially among those who rate their health as poor [3], indicating that adjudications of self-rated health encompass more than objective health information. As noted by Jylha (2009), self-rated health is “produced in a cognitive process that is inherently subjective” (p. 314) and involves evaluations of the presence and meaning of diverse health-related symptoms, experiences, and expectations. The outcome of this cognitive process is an assessment of health that predicts mortality risk beyond the risk attributable to known disease conditions or other health factors. Indeed, one longitudinal population study showed a dose-dependent relationship between self-rated health and mortality over a 30-year follow-up period, an association not easily explained by clinical risk factors or medical history [5]. In prior work, we showed that positive psychological functioning—specifically, purpose in life—moderated the association between chronic medical conditions and biological risk factors related to health. Specifically, circulating levels of inflammatory proteins increased linearly with a greater number of chronic conditions, but this increase was significantly less steep in those with greater purpose in life compared to those with less [6]. We now extend this earlier work to determine whether purpose in life also moderates the relationship between subjective health perceptions and mortality. 

Purpose in life—defined here as the extent to which individuals feel their lives are characterized by meaningful purpose and direction—has been extensively linked to diverse aspects of health [7]. In community-dwelling middle-aged and older adults, for example, a strong sense of purpose in life was cross-sectionally [8] and prospectively [9] associated with better self-rated health. Greater sense of purpose has also been shown to predict lower mortality risk in large population surveys, like the Midlife in the United States (MIDUS) study [10], the Health and Retirement Study (HRS) [11,12], and the English Longitudinal Study of Aging [13], and in community samples of healthy older adults [14]. Recent meta-analyses of studies involving tens of thousands of people reported that greater purpose in life was associated with significantly better self-rated health [15] and a significantly reduced risk of all-cause mortality [16]. Of relevance to the current analysis are studies showing the moderating effects of purpose in life. In the context of socioeconomic adversity [17] or multimorbidity [6], for example, those with a greater sense of purpose had lower levels of inflammation than those with less purpose, and in older adults with high levels of dementia-related brain pathology, cognitive function was better preserved in those with a greater sense of purpose [18]. Given these multiple lines of research, we hypothesize that greater purpose in life will be an independent predictor of mortality and will buffer the association between poor subjective health and a greater probability of mortality.

We also examine potential racial/ethnic differences in these associations, motivated by research suggesting that minoritized populations, especially Black people in the United States, may not benefit from purportedly health-promoting experiences and exposures to the same extent as White people for reasons of chronic systemic discrimination [19]. While educational attainment, for example, is broadly associated with gains in health, this is less true for Black Americans [20,21,22], although there have been recent improvements in this disparity [21], particularly among those earning educational credentials [23]. Where racial/ethnic differences in psychological well-being have been studied, Black respondents on average report comparable or higher levels of purpose in life compared to White respondents [24,25], and while at the population level Black people typically have worse subjective health and higher rates of mortality than White people on average [19], higher levels of flourishing broadly [26] and purpose in life specifically [25] are associated with reduced mortality in Black and White populations alike. In contrast, results from a recent experimental study showed higher levels of positive psychological functioning (positive mood, self-esteem, and self-acceptance) did not protect against respiratory infections to the same degree in Black as in White people [27]. Collectively, these studies support an examination of potential racial/ethnic differences in key associations, especially the extent to which Black respondents in particular show potential benefits from greater purpose in life. 

We use data from HRS and MIDUS, two large longitudinal studies of middle-aged and older adults, to examine the moderation of the well-established link between self-rated health and longevity by purpose in life. While HRS has a larger number of participants, MIDUS offers a five-decade age range (34–84 years of age) and thus the opportunity to test our hypotheses across a larger span of adulthood. We also examine potential racial/ethnic variability in these associations among Black, White, and Hispanic respondents in HRS and between Black and White respondents in MIDUS. 

## 2. Materials and Methods

### 2.1. Participants

Data for the current study are from two sources: the Health and Retirement Study (HRS) and the Midlife in the United States (MIDUS) study. 

#### 2.1.1. HRS

HRS is a multistage probability survey of adults aged 50 years and older, with oversampling of Black and Hispanic Americans and Florida residents. Core data (e.g., demographic characteristics, subjective health ratings) have been collected every 2 years since the study’s inception in 1992. We use data from 2006 and 2008, the first waves in which data on purpose in life were collected (half the sample completed the questions in 2006 and the other half in 2008). Variables from the Core Survey were from the same year that purpose-in-life data were collected. Complete data was available for 13,671 participants. 

#### 2.1.2. MIDUS

MIDUS is a national survey of the physical and mental health of middle-aged and older adults, begun in 1995–1996 (*n* = 7108), with two follow-up waves of data collection in 2004–2006 and 2013–2014. Data for the present study are from the 2004–2006 data collection (MIDUS 2). Mortality-adjusted retention was 75% from MIDUS 1 to MIDUS 2. A new sample of predominantly (97%) Black residents of Milwaukee County, WI (*n* = 592) was included in the MIDUS 2 sample. The greater racial diversity in the MIDUS 2 sample was the primary rationale for using data from MIDUS 2. All respondents completed telephone interviews and self-administered questionnaires in all three waves. Complete data was available for 4226 participants.

The response scales used to assess purpose in life differed between MIDUS and HRS, so models were estimated separately for each study. 

### 2.2. Measures

#### 2.2.1. Mortality

Mortality data on HRS participants through the end of 2018 was determined from exit interviews with next-of-kin and linkage to the National Death Index. Mortality data on MIDUS participants through March 2018 was determined from linkage to the National Death Index, sample maintenance procedures, and mortality closeout interviews. For the purposes of this study, the year of death was used as an indicator of mortality, and all deaths regardless of specific cause were included in analyses (all-cause mortality). Mortality was assessed using a dichotomous variable (1 = deceased).

#### 2.2.2. Purpose in Life

Purpose in life was assessed using the 7-item subscale from the Ryff Psychological Well-Being scales [28,29]. Participants responded to items such as “I have a sense of direction and purpose in life” and “My daily activities often seem trivial and unimportant to me” (reverse-scored). In MIDUS, participants responded using a 7-option Likert scale ranging from 1 (Strongly Disagree) to 7 (Strongly Agree), with 4 (Neither Agree nor Disagree) being the midpoint. In HRS, the Likert scale had six options ranging from 1 (Strongly Disagree) to 6 (Strongly Agree), with no mid-point option. Reliabilities on these scales for the analytical samples were 0.69 for MIDUS and 0.74–0.75 for the 2006 and 2008 HRS waves, respectively.

#### 2.2.3. Self-Rated Health

Participants rated their health using a single item. In HRS, participants were asked, “Would you say your health is excellent, very good, good, fair, or poor?” In MIDUS, participants were asked, “In general, would you say your physical health is excellent, very good, good, fair, or poor?” In both studies, responses were scored on a 1–5 scale, ranging from 1 (Poor) to 5 (Excellent). 

#### 2.2.4. Race

Among those in the analytical sample from HRS, 8% identified as Hispanic, 79% identified as non-Hispanic White, and 13% identified as non-Hispanic Black. All three racial/ethnic groups were included in analyses of HRS data. As 95% of the analytical sample from MIDUS was identified as non-Hispanic Black (12%) or non-Hispanic White (83%), analyses included only these two racial groups. 

#### 2.2.5. Covariates

The selection of covariates centered on potential confounding by demographic and health characteristics. A continuous variable for age and a dichotomous variable for sex (1 = female) were included in all analyses. Educational attainment in HRS was a continuous variable based on years of education. For MIDUS, a 3-level categorical variable where 1 = high school or GED; 2 = 2-year college degree or some of a 4-year degree; and 3 = 4-year college degree or more was used. Household wealth in the HRS sample was assessed by a continuous variable for total assets (e.g., real estate, businesses, investments, and savings) less total debt (mortgages and other debt obligations). Due the skewed distribution, the variable was cube-root transformed for analysis. In MIDUS, participants were asked if they would have money left over after liquidating all of their household assets and paying off all debt. Response options were “Would have money left over; Would still owe money; and Debts would just about equal assets” and were dummy-coded for analyses with “Debts would just about equal assets” as the reference category. This variable was previously used to examine links between wealth and longevity in MIDUS [30]. Health issues in both studies were assessed using an aggregate measure of seven chronic conditions for which participants had received a physician’s diagnosis and/or treatment: hypertension, heart disease, cancer, stroke, arthritis, lung disease, and diabetes (1 = yes; range 0–7). 

### 2.3. Analytical Approach

Analyses were completed using Stata 18 (StataCorp, College Station, TX, USA). Tests of the proportional hazards assumption showed that the assumption was violated, making the use of Cox Proportional Hazards models inappropriate. Separate logistic regression models were used to estimate the probability of mortality in the HRS and MIDUS samples. As the MIDUS sample includes twins and siblings, and for consistency across analyses, robust standard errors were used in all models. Models for the full samples and samples stratified by race are shown. Results are average marginal effects (AMEs), which indicate on average the effect on mortality probability of a one-unit increment in the independent variable (or change in category for categorical variables). As they indicate changes in probability rather than changes in a multiplicative function such as odds ratios, average marginal effects (AME) are more easily interpreted [31]. 

To test for moderation of self-rated health by purpose in life, interaction terms (self-rated health X purpose in life) were added to the models, although tests of the interaction involved additional steps. In linear models, it is customary to consider an interaction coefficient as evidence of moderation. However, in logistic regression models, where the distribution of the dependent variable is inherently non-linear, the interaction term cannot easily be directly interpreted. In fact, both the direction and significance level of the interaction term in a logistic regression model can differ from the actual underlying effect [32,33]. For this reason, it is instead advisable to use tests of the predicted probabilities to determine whether a moderating effect exists across particular levels of interest [34], in this case the difference in mortality probability for those rating their health as “poor” vs. “excellent.” We first examined the difference in mortality probability between “poor” and “excellent” self-rated health for those who were one standard deviation above (+1 SD) and one standard deviation below (−1 SD) the sample mean of purpose in life. These are tests of “first differences” [34], indicating whether the mortality probabilities for differences in self-rated health at the different ‘levels’ of purpose in life were significantly different from 0. We then determined whether these group differences (+1 SD vs. −1 SD) were significantly different from one another (referred to as “second differences”). The second difference tests the potential interaction, that is, the extent to which the relationship between self-rated health and mortality varies across the different ‘levels’ of purpose in life. 

## 3. Results

Descriptive statistics for HRS and MIDUS—both the full and stratified samples—are shown in Table 1 and Table 2, respectively. Among the different racial/ethnic groups in HRS (Table 1), White participants were significantly older and had greater male representation, more years of education, greater household wealth, and greater self-rated health. Black participants had comparatively more chronic conditions and higher ratings on purpose in life. Hispanic participants had the lowest mortality rate and the fewest chronic conditions. Results were similar for MIDUS respondents (Table 2), except that White and Black participants had comparable ratings on purpose in life. Between the two studies, HRS participants were older on average and had more chronic conditions.

Correlational analyses showed that self-rated health and purpose in life were significantly but moderately correlated in the full HRS sample (*r* = 0.29, *p* < 0.001), with a stronger correlation in White (*r* = 0.31, *p* < 0.001) and Hispanic (*r* = 0.31, *p* < 0.001) participants than in Black participants (*r* = 0.25, *p* < 0.001). The correlation was almost identical in the MIDUS sample (*r* = 0.28, *p* < 0.001), and there was also a stronger correlation in White participants (*r* = 0.30, *p* < 0.001) than in Black participants (*r* = 0.20, *p* < 0.001). 

Logistic regression analyses estimating probability of mortality for the full HRS sample showed that one-point increments in self-rated health (AME = −0.06, *p* < 0.001) and purpose in life (AME = −0.03, *p* < 0.001) were associated with significantly lower probability of mortality, adjusted for demographic and health covariates. Analyses stratified by race/ethnicity showed similar results. Covariates were significantly associated with mortality probability in the expected directions, with the exception of years of education, where each additional year of education predicted a 0.01 lower probability of mortality for White people only (see Table A1).

The results of interaction analyses with HRS data are presented in Table 3. On average for the full sample, self-rated health had a weaker effect on mortality probability for those scoring higher on purpose in life (Δ_+1 SD_ = 0.23; *p* < 0.001) than for those scoring lower (Δ_−1 SD_ = 0.31; *p* < 0.001; second difference = 0.08, *p* < 0.05), indicating a significant interaction. Among those rating their health as poor, the probability of dying during the follow-up period was 17% lower for those with low scores on purpose in life compared to those with higher scores (0.44 vs. 0.53). Put another way, the slope of the line for higher purpose in life (+1 SD) was not as steep (i.e., a weaker association with self-rated health) as the line for lower (−1 SD) purpose, and the magnitude and statistical significance of this difference in slope are given by the second difference test. The interaction results suggested that rating one’s health as poor was more detrimental (in terms of mortality) for those who also reported low purpose in life, compared to those with high purpose.

Analyses stratified by race showed that results were generally consistent across White, Black, and Hispanic participants (see Figure 1). Among White participants, self-rated health had a weaker effect on mortality probability for those scoring higher on purpose in life (Δ_+1 SD_ = 0.26; *p* < 0.001) than for those scoring lower (Δ_−1 SD_ = 0.32; *p* < 0.001; second difference = 0.06, *p* < 0.05). Among Black participants, self-rated health also had a weaker effect on mortality probability for those scoring higher on purpose in life (Δ_+1 SD_ = 0.17; *p* < 0.001) than for those scoring lower (Δ_−1 SD_ = 0.31; *p* < 0.001; second difference = 0.13), and while the interaction was not statistically significant, the magnitude of the second difference was more than twice that for White participants. Results for Hispanic participants showed that while self-rated health had an effect on mortality probability for those scoring lower on purpose in life (Δ_−__1 SD_ = 0.24; *p* < 0.001), it was unrelated to mortality probability for those scoring higher (Δ_+1 SD_ = 0.05; *p* = 0.50; second difference = 0.19, *p* < 0.05). Among those rating their health as poor, the probability of mortality during the follow-up period was 13%, 26%, and 39% lower among White, Black, and Hispanic participants, respectively, for those with high scores on purpose in life compared to those with lower scores. 

Logistic regression analyses estimating probability of mortality for the full MIDUS sample showed that 1-point increments in self-rated health (AME = −0.04, *p* < 0.001) and purpose in life (AME = −0.02, *p* < 0.001) were associated with significantly lower probability of mortality, adjusted for demographic and health covariates. Analyses stratified by race/ethnicity showed similar results for self-rated health, but purpose in life was not associated with reduced mortality in Black respondents. Covariates were significantly associated with mortality probability in the expected directions, with the exception of educational attainment. Greater educational attainment was not associated with mortality probability in the full sample, while some college (compared to a high school degree or GED) predicted a lower (0.07) mortality probability in Black respondents, but an increased (0.03) mortality probability in White respondents (see Table A2).

The results of interaction analyses with MIDUS data are shown in Table 4. On average for the full sample, self-rated health had a weaker effect on mortality probability for those scoring higher on purpose in life (Δ_+1 SD_ = 0.19; *p* < 0.001) than for those scoring lower (Δ_−1 SD_ = 0.28; *p* < 0.001; second difference = 0.09, *p* < 0.05), indicating a significant interaction. Among those rating their health as poor, the probability of dying during the follow-up period was 31% lower for those with high scores on purpose in life compared to those with lower scores (0.25 vs. 0.36). 

Analyses stratified by race showed that among White respondents, self-rated health had a weaker effect on mortality probability for those scoring higher on purpose in life (Δ_+1 SD_ = 0.16; *p =* 0.001) than for those scoring lower (Δ_−1 SD_ = 0.28; *p* < 0.001; second difference = 0.12 *p* < 0.01), indicating a significant interaction. In contrast, among Black participants, there was little difference in the strength of the association between self-rated health and mortality based on purpose in life (Δ_+1 SD_ = 0.33; *p* < 0.01; Δ_−1 SD_ = 0.28; *p* < 0.01; second difference = 0.05, *ns*). The results of interaction analyses with MIDUS data are presented in Table 4 and shown graphically in Figure 2.

Subjective health is a robust predictor of longevity over and above observable clinical information about individual health. Drawing from a growing literature on the salubrious effects of psychological well-being, we tested the hypothesis that purpose in life may at least partially moderate the association between subjective health and mortality. The overall results supported the hypothesis: in two large national samples of middle-aged and older adults, the relationship between self-rated health and mortality was significantly more modest for people who reported greater purpose in life than for those reporting less purpose. In other words, greater purpose in life buffered the relationship between poorer self-rated health and a greater probability of mortality. On average, greater purpose in life (+1 SD vs. −1 SD) reduced the probability of mortality by 0.11 (HRS) and 0.09 (MIDUS) among those rating their health as poor. These effects exceed the change in predicted mortality probability for each additional chronic condition (0.03 in both HRS and MIDUS) and are comparable to the effect associated with an age difference of 5.5 years in HRS and 9 years in MIDUS. This was true even though mortality rates differed markedly between HRS and MIDUS, likely due to the older age of the HRS sample. These results suggest that greater purpose in life predicts meaningful reductions in mortality associated with poorer self-rated health among middle-aged and older adults.

Self-rated health is the result of a cognitive process that takes into account multiple factors, including interpretations of health-related information and symptoms that are shaped by cultural definitions of health; norms and expectations related to age as well as intra- and interpersonal comparisons; and cultural and personal influences on the interpretation of options to respond to the relevant question [3]. The reasons that it is such a powerful predictor of mortality beyond its associations with clinical conditions, however, are less clear, although they arguably involve variance in biological processes that is not captured by known chronic illnesses or risk factors [3]. Earlier work showed that purpose in life is one element that informs an individual’s subjective health assessment, with greater purpose predicting better self-rated health [8,9]. The magnitude of the zero-order correlations between purpose and self-rated health in both MIDUS and HRS was comparable to studies of other populations, such as the Hawaii Study of Personality and Health (0.29; [8]). However, this association does not explain the moderation effects observed here, as there were participants in MIDUS and HRS who rated their health as “poor” while scoring high on purpose in life; that is, high purpose ratings did not translate into better subjective health for some. The current results therefore suggest both that purpose in life is independently associated with mortality and that it reflects or predicts one or more processes that directly affect longevity but that are not represented in adults’ subjective health ratings. 

What might these processes be? One possibility is differences in the specific health factors linked to self-rated health and purpose in life. A “poor” subjective assessment of health in older adults is most likely to be driven by the presence of chronic conditions and functional impairment [4]. However, the presence of chronic conditions and functional limitations is not a barrier to higher levels of purpose in life [6], and older adults with a greater sense of life purpose are more likely to engage in health-promoting behaviors that can increase longevity, including physical activity participation [8,35,36], use of preventive services [37], and getting better sleep [8,38,39]. Greater purpose is also linked to more favorable profiles of biological function, including reduced activation of genes related to inflammation [40,41] and lower circulating levels of inflammatory proteins [6], that may not be captured by self-rated health and that predict mortality [42]. Purpose in life and the purposeful activities it promotes [43] may thus contribute to longevity even when the objective metrics of health on which subjective health ratings largely rest are more consistent with a greater mortality risk. This perspective is consistent with theoretical models, such as resource substitution [44], that describe the processes by which strengths in one domain compensate for deficits in another. If self-rated health and purpose in life can be considered resources for attaining longevity, for example, greater purpose in life would be expected to have a larger impact on longevity for those with poor self-rated health than for those who consider their health to be excellent. 

We also examined the variability in these associations by race/ethnicity. With the exception of Black MIDUS participants, greater purpose in life was associated with greater longevity in all respondents and weakened the link between poorer self-rated health and greater mortality. The magnitude of the second difference for Black HRS participants supports this conclusion, despite the lack of statistical significance. Our interpretation aligns with Willroth and colleagues (2021), who concluded that changes in purpose in life over time predicted health outcomes similarly in Black and White respondents, despite statistically non-significant effects among Black respondents [45]. More broadly, our finding that greater purpose in life was associated with greater longevity is consistent with prior studies showing that purpose in life predicts reduced mortality risk [25] and reduced risk of dementia [46] similarly in White and Black participants; moreover, some research shows greater protection against cognitive decline among Black participants specifically [47], similar to the results in the HRS sample. In contrast, purpose in life did not, on average, predict mortality among Black MIDUS respondents, and there was little evidence of an interaction with self-rated health. One potential explanation for these differential results is the relatively smaller number of Black MIDUS respondents (*n* = 535), which may yield less reliable estimates. That said, the results also show substantial variability around those predicted mortality estimates, suggesting meaningful heterogeneity in this population. The descriptive data show large socioeconomic heterogeneity among the Black MIDUS respondents that might affect key associations in this study. For example, prior research has shown that the strength of the association between self-rated health and mortality varies with socioeconomic status [48]. An examination of the heterogeneity within this specific population is beyond the scope of the current study but an important focus for future efforts. 

There are a number of limitations that should temper the interpretation of the current results. Chronic conditions were reported by participants rather than confirmed by medical records and are therefore subject to recall inaccuracies that could inflate the association between self-rated health and mortality probability. Low socioeconomic status (SES) is underrepresented in both samples, and SES has been shown to moderate the association between self-rated health and mortality [48]. Finally, Black respondents in MIDUS are mostly (71%) from a regional sample of residents of Milwaukee County, Wisconsin, and therefore may not accurately reflect the Black population of the US. The regional nature of the Black MIDUS sample may at least partially account for the lack of association between purpose in life and mortality. Against these limitations, there are considerable strengths. MIDUS and HRS are two large, well-established national longitudinal studies that are widely used in population research, and the current results are strikingly similar in both despite a large difference in mean age. The measures of self-rated health and purpose in life are similarly widely used and reliable. The follow-up periods for mortality estimates were long, approximately 12–14 years in both studies, increasing confidence in the robustness of these observations.

## 4. Conclusions

The results of this study suggest that a strong sense of purpose in life is linked to greater longevity and may protect against the detrimental effects of poor subjective health. These results join a growing literature citing the potential health benefits of purpose in life specifically and positive psychological functioning more broadly. Importantly, there are a growing number of clinic- and community-based interventions that have been shown to promote purpose in life [49,50], offering the potential to improve health and possibly longevity in middle-aged and older adults.

## Figures and Tables

**Figure 1 ijerph-20-06171-f001:**
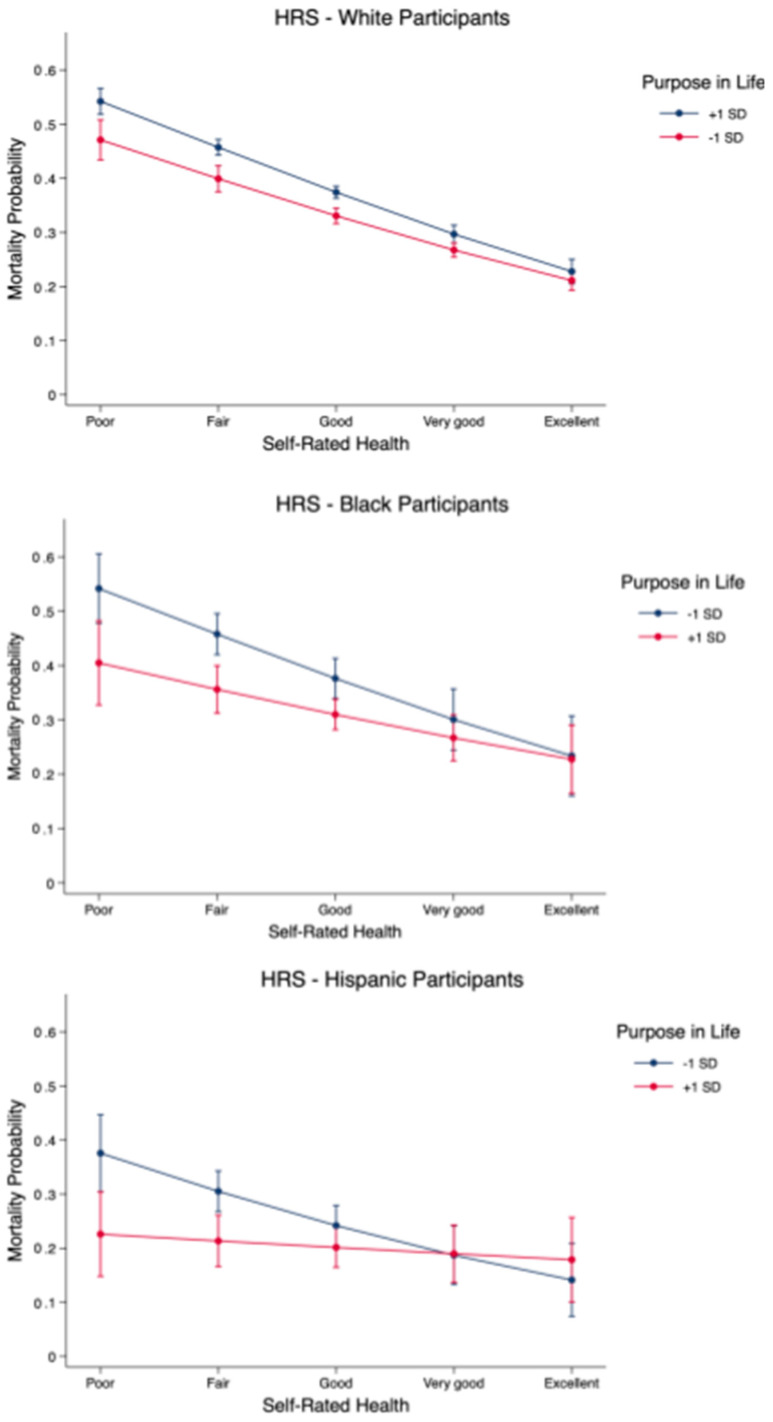
Probability of mortality in HRS across ratings of self-rated health and +1/−1 SD on purpose in life, stratified by race.

**Figure 2 ijerph-20-06171-f002:**
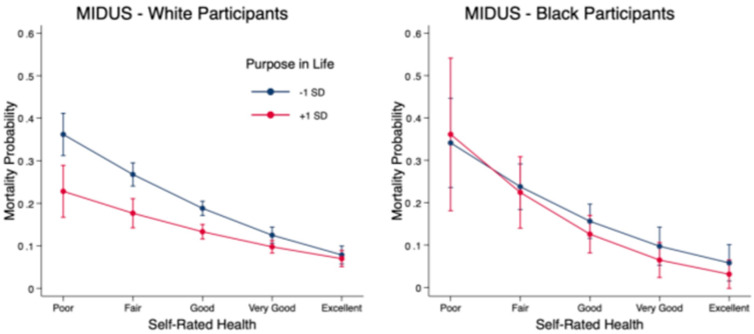
Probability of mortality in MIDUS across ratings of self-rated health and +1/−1 SD on purpose in life, stratified by race.

**Table 1 ijerph-20-06171-t001:** Descriptive statistics for the HRS sample.

Variable	Full Sample (*n* = 13,671)			White Participants (*n* = 10,791)			Black Participants (*n* = 1812)			Hispanic Participants (1199)		
	Mean (SD)	Range	%	Mean (SD)	Range	%	Mean (SD)	Range	%	Mean (SD)	Range	%
Deceased			33.9			34.5			35.9			25.4
Self-Rated Health	3.2 (1.1)	1–5		3.3 (1.1)	1–5		2.8 (1.1)	1–5		2.7 (1.1)	1–5	
Purpose in Life	4.5 (1.0)	1–6		4.5 (1.0)	1–6		4.7 (1.0)	1–6		4.4 (1.0)	1–6	
Age	67.5 (10.4)	26–104		68.0 (10.3)	30–101		66.2 (10.0)	26–99		64.5 (10.6)	31–104	
Sex (1 = female)			59.3			58.0			65.7			61.7
Education (years)	12.6 (3.1)	0–17		13.1 (2.6)	0–17		11.7 (3.2)	0–17		9.3 (4.6)	0–17	
Chronic conditions	2.1 (1.4)	0–7		2.0 (1.4)	0–7		2.3 (1.4)	0–7		1.9 (1.3)	0–6	
Household wealth (Median; $1000s)	$213.8 ($1262.2)	−$2199.4–$41,664.8		$285.0 ($1402)	−$2199.4– $41, 664.8		$51.0 ($332.0)	−$120.8–$4476.0		$70.25 ($556.0)	−$550–$11,407.7	

**Table 2 ijerph-20-06171-t002:** Descriptive statistics for the MIDUS sample.

Variable	Full Sample (*n* = 4226)			White Participants (*n* = 3691)			Black Participants (*n* = 535)		
	Mean (SD)	Range	%	Mean (SD)	Range	%	Mean (SD)	Range	%
Deceased			15.0			14.9			15.9
Self-Rated Health	3.5 (1.0)	1–5		3.6 (1.0)	1–5		3.1 (1.1)	1–5	
Purpose in Life	5.5 (1.0)	1–7		5.5 (1.0)	1–7		5.4 (1.1)	2–7	
Age	56.0 (12.4)	30–85		56.3 (12.4)	32–84		53.4 (11.7)	30–85	
Sex (1 = female)			56.2			55.1			64.5
Education									
High School/GED			35.1			32.6			52.5
Some College			28.9			28.9			29.9
College or more			36.0			38.7			17.8
Chronic conditions	0.8 (0.9)	0–7		0.8 (1.0)	0–7		1.0 (1.1)	0–5	
Household wealth									
Would still have money			77.9			82.8			43.3
Debts equal assets			8.5			7.3			17.3
Would still owe			13.6			10.0			39.4

**Table 3 ijerph-20-06171-t003:** Tests of first and second differences in HRS. Model-derived estimates of predicted mortality probability at ratings of “poor” and “excellent” for self-rated health and +/− 1 SD for purpose in life are shown. All estimates are adjusted for demographic and health covariates.

**Full Sample (*n* = 13,671)**			**Self-Rated Health**		**Difference**
			**Poor**	**Excellent**	
First Differences	Purpose in Life	+1 SD	0.44	0.21	0.23, *p* < 0.001
		−1 SD	0.53	0.22	0.31, *p* < 0.001
Second Difference	(−1 SD)–(+1 SD)		0.31–0.23 = 0.08, *p* < 0.01		
**White participants (*n* = 10,791)**			**Self-Rated Health**		**Difference**
			**Poor**	**Excellent**	
First Differences	Purpose in Life	+1 SD	0.47	0.21	0.26, *p* < 0.001
		−1 SD	0.54	0.22	0.32, *p* < 0.001
Second Difference	(−1 SD)–(+1 SD)		0.32–0.26 = 0.06, *p* < *0*.05		
**Black Participants (*n* = 1731)**			**Self-Rated Health**		**Difference**
			**Poor**	**Excellent**	
First Differences	Purpose in Life	+1 SD	0.40	0.23	0.17, *p* < 0.01
		−1 SD	0.54	0.23	0.31, *p* < 0.001
Second Difference	(−1 SD)–(+1 SD)		0.31–0.17 = 0.13, *p* = 0.11		
**Hispanic Participants (*n* = 1149)**			**Self-Rated Health**		**Difference**
			**Poor**	**Excellent**	
First Differences	Purpose in Life	+1 SD	0.23	0.18	0.05, *p* = 0.50
		−1 SD	0.38	0.14	0.24, *p* < 0.001
Second Difference	(−1 SD)–(+1 SD)		00.24–0.05 = 0.19, *p* < 0.05		

**Table 4 ijerph-20-06171-t004:** Tests of first and second differences in MIDUS. Model-derived estimates of predicted mortality probability at ratings of “poor” and “excellent” for self-rated health and +/− 1 SD for purpose in life are shown. All estimates are adjusted for demographic and health covariates.

**Full Sample (*n* = 4226)**			**Self-Rated Health**		**Difference**
			**Poor**	**Excellent**	
First differences	Purpose in Life	+1 SD	0.25	0.06	0.19, *p* < 0.001
		−1 SD	0.36	0.08	0.28, *p* < 0.001
Second difference	(−1 SD)–(+1 SD)		0.28–0.19 = 0.09, *p* < 0.05		
**White participants (*n* = 3691)**			**Self-Rated Health**		**Difference**
			**Poor**	**Excellent**	
First Differences	Purpose in Life	+1 SD	0.23	0.07	0.16, *p* = 0.001
		−1 SD	0.36	0.08	0.28, *p* < 0.001
Second Difference	(−1 SD)–(+1 SD)		0.28–0.16 = 0.12, *p* < 0.01		
**Black Participants (*n* = 535)**			**Self-Rated Health**		**Difference**
			**Poor**	**Excellent**	
First Differences	Purpose in Life	+1 SD	0.36	0.03	0.33, *p* < 0.01
		−1 SD	0.34	0.06	0.28, *p* < 0.01
Second Difference	(−1 SD)–(+1 SD)		0.28–0.33 = −0.05, *p* = 0.67		

## Data Availability

The data used for this study are publicly available: HRS: https://hrs.isr.umich.edu/about (accessed on 15 October 2022). MIDUS: https://midus.colectica.org/ (accessed on 15 October 2022).

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
