# Peer review of "Self-Rated Health and Mortality: Moderation by Purpose in Life"

_ijerph, 2023, doi:10.3390/ijerph20126171_

Round 1

Reviewer 1 Report

Summary

·         Harnessing data from MIDUS and HRS, the authors evaluated if purpose in life moderated the relationship between self-rated health and mortality. The authors also evaluated potential differences in these associations by race/ethnicity. Results showed that purpose in life moderated the self-related health and mortality association, but weaker effects were observed for Black respondents. This was an interesting and well written paper, below are some additional factors to consider which might even further enhance this paper. 

Feedback

·         Introduction

o   The introduction was well-written and interesting to read.

·         Methods

o   Did the authors consider controlling for income or wealth? These might be potentially important confounders to consider.

o   I think I know why the authors used MIDUS 2 as the baseline wave, but it would be helpful to state this explicitly.

o   The authors stated that they are using average marginal effects (AMEs) because the estimates are more easily interpreted. I thank the authors for the using the approach. However, when discussing results from AMEs analyses the language used is often a variation of “were associated with significantly lower probability of mortality.” Could the authors interpret the AMEs results in ways that are more easily understood? This would help readers grasp the results more quickly, not if a result is significant/not significant but also the magnitude of the result.

·         Results

o   In HRS analyses, the estimates showed that greater purpose in life may lower the likelihood of mortality in individuals who perceive their health to be poor, but that these effects were weaker in Black respondents. However, when I compare the first differences and second difference estimates in Black participants, against the estimates from White participants and the Full Sample, the estimates are actually very similar. Is it possible that weaker effects were observed in Black participants due to a lower sample size/power? If so, could the authors address this idea in the limitations section?

o   In HRS analyses, why do you think the interesting results emerged in the Hispanic participants?

o   In MIDUS analyses, purpose in life was not associated with reduced mortality in Black respondents. Why do you think this might be?

o   It was helpful to see both the Tables and Figures, as they both conveyed insights in different ways.

·         Discussion

o   It would be helpful to situate these findings more in the context of prior work that has examined how self-rated health and mortality, as well as purpose in life and mortality, might be moderated by race/ethnicity. Doing so might help provide more explanation for these intriguing findings, as there are many that the authors uncovered in this very interesting study.

o   Are there any interventions, or promising interventions, that might impact levels of purpose in life? If so, could the authors discuss this literature, even briefly, in the Discussion section?   

Reviewer 2 Report

The current manuscript sought to consider the potential moderators of associations between self-rated health and mortality risk, focusing on sense of purpose and race/ethnicity. Researchers replicated the previous evidence in MIDUS and HRS that self-rated health and purpose predict longevity, and found modest support that the two factors may interact in this prediction. Although these findings are potentially valuable to the literature, the authors failed to provide a strong rationale toward that end, which left the manuscript unmotivated and with the feeling of a post hoc explanation.

First, with respect to self-rated health, the narrative centers on how this variable is a “cognitive process” shaped by several factors. Certainly, self-rated health is not a purely objective indicator of health, and much has been written about what this variable actually captures. However, the authors lean far too heavily into this subjectivity issue when attempting to motivate looking at moderators. Though it has its flaws, the largest influence on self-rated health is clearly someone’s actual health status. Accordingly, the most obvious reason why sense of purpose would serve as a moderator is that for people in great health, psychosocial variables will generally play less of a role. This point aligns with the resource substitution (e.g., Ross & Mirowsky, 2006; Social Science & Medicine) – for individuals with fewer resources (health, income, etc.), psychosocial variables will play a bigger role, whereas those individuals with high resources, psychosocial variables will be less relevant for health and wellbeing. This account makes much more sense for motivating interactions between sense of purpose and self-rated health, rather than attempting to posit how sense of purpose would play a role in the cognitive processes involved in self-rated health. I would strongly encourage the authors to revisit the introduction and discussion, and provide a better explanation for considering moderation by sense of purpose. Currently, the authors have failed to properly portray self-rated health, by moving too far onto the side of “we don’t know why self-rated health would be important” rather than acknowledging it for the biased, though fairly valid indicator it truly is.

Second, the rationale for examining racial/ethnic differences was unfortunately even less clear. The authors rely on the fact that race is associated with purpose, health, and resources, which in and of itself does not suggest moderation by race/ethnicity. Multiple studies have shown that sense of purpose appears to operate similarly across racial/ethnic groups (see e.g., Willroth et al., 2022; J of Psychosomatic Medicine), which are missing in the introduction. Overall, this section of the introduction needed much greater length and attention, and I would encourage the authors to revisit the purpose literature to better couch this work in the existing literature. Furthermore, this section again suggested the potential for a resource substitution account, as the authors are intentionally invoking resources as a possible reason why these variables may play differing roles in predicting mortality.

Beyond the need for much clearer rationale in the introduction, the methods and results required more explanation as well. For instance, it was unclear why the authors chose logistic regressions here, rather than the more widely accepted Cox regression models. Cox analyses would have allowed for clearer tests of interaction factors instead of the odd average marginal effects strategy, which problematically requires dichotomization of both primary variables. Several of the papers references by the authors in fact have employed this analytic strategy in their papers with HRS and MIDUS data. In addition, the interpretation of the AMEs was very problematic, as the authors focused on statistical significance as the benchmark for interpretation. As an example from Table 3, the second difference was over twice as large for Black participants than White participants, and Figure 1 also shows the interaction was much clearer for Black participants. However, this isn’t recognized by the authors, who needed to focus more on effect size than significance throughout the document. 

In sum, the reader is left very unclear whether this paper provides a contribution to the literature. To start, the authors need a much better motivation for the work and could easily double or triple the length of the introduction, in order to better couch this work. In addition, the findings are potentially suspect given the analytic strategy chosen, and leaves one wondering whether the interaction findings would differ had the authors employed a more traditional strategy. 

The document in the whole requires a thorough proofread. Several paragraphs, particularly in the introduction, included articles that did not flow naturally from one claim to the other.

Reviewer 3 Report

The reviewer examined the submitted paper titled “Self-rated health and mortality: Moderation by purpose in life,” with tremendous interest. The study clarified that the positive association between self-rated health and mortality was moderated by purpose in life, based on analytical results using data from two large national longitudinal surveys. Additionally, the study indicated that such positive influences of purpose in life on mortality differed between races/ethnicities (particularly White and Black respondents). The reviewer believes that the findings presented in the study may contribute to the literature in public health studies.

Specifically, the study critically argued the contributions of previous studies related to the association between subjective health and longevity and effectively formulated their research questions. Simultaneously, the authors effectively used panel data-sets (HRS and MIDUS) and adopted a suitable analytical method to examine their hypotheses. Consequently, the analytical results presented in the study are persuasive for the reviewer. Therefore, the reviewer believes that the submitted paper has reached a sufficient level for publication.

However, the reviewer has a minor concern regarding the authors’ interpretation of the analytical results, which relates to the authors’ conclusion. The reviewer is afraid that the authors’ interpretations might confuse the readers’ understanding of the author’s study. If the authors adequately address this point, the submitted paper would be significantly improved and potential critiques could be avoided. If the authors believe that they do not need to revise the submitted paper, they should provide adequate reasons.

According to Table 3, the second difference for White participants (n=10,791) was statistically significant (.34-.28=.06, p<.05). However, the second difference for Black participants (n=1,731) was not statistically significant (.34-.21=.13, p=.12). However, the value of the second difference for Black participants was greater than that of the second difference for White participants (.13 > .06). Therefore, the reviewer suspects whether the results of the statistical tests for the second difference depend only on the difference in the number of participants between the two racial groups. Thus, it is assumed that, as the number of White participants (n=10,791) is sufficiently large, the statistical tests of the second difference for White participants could be passed. Similarly, it is assumed that as the number of Black participants (n=1,731) is not sufficiently large, the statistical tests of the second difference for Black participants could not be passed. If this is true, the reviewer believes that the authors should not insist that there is a significant difference in the effects of purpose in life on the association between self-rated health and mortality between the two racial groups. Probably, if the number of Black participants was the same as that of White participants, the statistical tests on the second difference for Black participants would be passed.

A similar problem was observed with the MIDUS (Table 4). In the MIDUS, the number of participants was also different between the White (n=3,691) and Black (n=1,731) groups. The reviewer suggests that the authors should not focus on the statistical significance of the second differences alone. He believes that the authors could consider the differences in effect size. Additionally, the authors may perform a robustness check using the Bootstrap method.

The reviewer thanks the authors for providing the opportunity to review the manuscript and sincerely expects that these comments will be useful for revising the manuscript.

Round 2

Reviewer 2 Report

Overall, the document is significantly improved over the previous draft, and I appreciate the authors' attention to the past reviewer comments. That said, two issues remain with the current document, which I think could be improved upon in another revision.

First, the authors have yet to provide a strong rationale for why they chose the logistic regression strategy. Again, they are not taking a typical strategy for examining mortality risk, which in turn yields a more complicated, less intuitive approach to testing interactions. All the authors provided in their response was to that they decided to stick with their current strategy, despite acknowledging it was less typical. Therefore, it would be valuable to include in the text a rational for the current analytic strategy.

Second, related, the interaction interpretations still are quite difficult to follow, in part given the uncommon approach. I do not usually suggest adding "discussion" to the results section, but this is a case where the authors may want to include a few sentences in each section to really describe the interaction results.

Two additional notes:

1) Personally, I would move the appendices into the text proper. The claims made by the authors in text cannot currently be supported by tables in the document, and the reader requires the additional ones to understand the discussion section. As such, it seems valuable to include the additional tables in the manuscript as well.

2) Related, the most likely explanation for the discrepant findings for Black participants in MIDUS is the sample size. Conducting a mortality analysis with only 535 participants is going to lead to unreliable estimates in most cases, particularly with intercorrelated predictors (as the authors have here). As such, I would soften claims regarding why purpose and other variables fail to predict in this case, and emphasize the lack of power once broken down by race.

N/A

Author Response

First, the authors have yet to provide a strong rationale for why they chose the logistic regression strategy.

At this reviewer’s suggestion, in preparing our first revision we estimated mortality using a Cox Proportional Hazards model with HRS data, and the results closely matched those from the original logistics models. However, after submitting the first revision we did some additional analyses and found that the proportional hazards assumption was violated making the results from the Cox model unreliable. For this reason, we present results from the logistic regression models. We have added the following text to the relevant section of the Method:

Analyses were completed using Stata 18 (StataCorp, College Station, TX). Tests of the proportional hazards assumption showed that the assumption was violated, making the use of Cox Proportional Hazards models inappropriate. (p. 8)

Second, related, the interaction interpretations still are quite difficult to follow, in part given the uncommon approach. I do not usually suggest adding "discussion" to the results section, but this is a case where the authors may want to include a few sentences in each section to really describe the interaction results.

We added the following two sentences to the Results section to try to further clarify the interaction results. This text follows the presentation of results for HRS analyses. We did not add the same text after the section for MIDUS data, as it seemed repetitive. We hope this addresses the reviewer’s remaining concern.

Put another way, the slope of the line for higher purpose in life (+1 SD) was not as steep (i.e., a weaker association with self-rated health) as the line for lower (-1 SD) purpose, and the magnitude and statistical significance of this difference in slope is given by the second difference test. The interaction results suggested that rating one's health as poor was more detrimental (in terms of mortality) for those who also reported low purpose in life, compared to those with high purpose. (p. 13)

Two additional notes:

  • Personally, I would move the appendices into the text proper. The claims made by the authors in text cannot currently be supported by tables in the document, and the reader requires the additional ones to understand the discussion section. As such, it seems valuable to include the additional tables in the manuscript as well.

We are open to this suggestion, but mindful of space limitations we leave the decision on where to place these materials to the editorial team.

  • Related, the most likely explanation for the discrepant findings for Black participants in MIDUS is the sample size. Conducting a mortality analysis with only 535 participants is going to lead to unreliable estimates in most cases, particularly with intercorrelated predictors (as the authors have here). As such, I would soften claims regarding why purpose and other variables fail to predict in this case, and emphasize the lack of power once broken down by race.

We agree with the reviewer on this point, and we have added the following text to the Discussion:

One potential explanation for these differential results is the relatively smaller number of Black MIDUS respondents (n=535) that may yield less reliable estimates. That said, the results also show substantial variability… (p. 20)